# Ensemble Learning-Based Alzheimer’s Disease Classification Using Electroencephalogram Signals and Clock Drawing Test Images

**DOI:** 10.3390/s25092881

**Published:** 2025-05-02

**Authors:** Young Jae Huh, Jun-ha Park, Young Jae Kim, Kwang Gi Kim

**Affiliations:** 1Department of Medicine, Yonsei University Wonju College of Medicine, Wonju 26426, Republic of Korea; chuh@yonsei.ac.kr; 2Department of Biomedical Engineering, College of Health Science, Gachon University, Incheon 21936, Republic of Korea; wnsgk1208@gachon.ac.kr; 3Medical Devices R&D Center, Gachon University, Gil Medical Center, Incheon 21565, Republic of Korea; kimyj10528@gmail.com; 4Department of Biomedical Engineering, College of Medicine, Gachon University, Incheon 21565, Republic of Korea; 5Department of Health Sciences and Technology, Gachon Advanced Institute for Health Sciences and Technology (GAIHST), Gachon University, Incheon 21999, Republic of Korea

**Keywords:** Alzheimer’s disease, clock drawing test, electroencephalogram, ensemble learning, machine learning

## Abstract

**Highlights:**

**Abstract:**

Ensemble learning (EL), a machine learning technique that combines the results of multiple learning algorithms to obtain predicted values, aims to achieve better predictive performance than a single learning algorithm alone. Machine learning techniques, including EL, have been applied in the field of medicine to assist in the clinical interpretation of specific diseases. Although neurodegenerative diseases, especially Alzheimer’s disease (AD), are of interest to clinicians and researchers due to their rapid increase in clinical cases, the application of EL in AD diagnosis has been relatively less attempted. In this research, we demonstrate that three machine learning algorithms, trained on an ensemble of electroencephalogram (EEG) and clock drawing test (CDT) feature data for an AD classification task, show improved AD detection accuracy compared to when either the EEG or CDT dataset is used independently. We also explore which feature contributes most to decision-making in AD and healthy control (HC) classification. In conclusion, the current study suggests that EL can be a novel clinical application of machine learning (ML) in the automated AD screening process.

## 1. Introduction

Dementia is an acquired and degenerative syndrome characterized by a general decrease in cognitive abilities. Alzheimer’s disease (AD) is the most common cause of dementia, accounting for 60–80% of cases [1]. Due to its close relevance to aging, AD has become relatively more prevalent compared to the past as the number and proportion of older people (65+ years) in the world’s population continue to rise [2,3,4]. In particular, the number of people living with clinical AD is projected to increase from 6.07 million in 2020 to 13.85 million in 2060 in the United States [5]. Therefore, developing efficient AD diagnostic tools is one of the major interests among clinicians and researchers. The chief problems the clinicians have brought up are its time-consuming nature and difficulty in clinical interpretation of the datasets from medical examinations such as electroencephalogram (EEG), magnetic resonance imaging (MRI), and mini-mental state examination (MMSE). Artificial intelligence (AI) techniques are state-of-the-art solutions that have been widely suggested to overcome such problems.

The importance of early AD detection has triggered the development of numerous cutting-edge diagnostic markers. Machine learning (ML) approaches have been widely used to construct automated environments for AD diagnosis. Previous studies have proposed that an ML-based approach to either EEG or CDT datasets could be used as an automatic diagnostic tool. Such automation helps clinicians save time and reduce bias caused by the clinician’s subjective judgment. In addition to conventional ML, some of the recent studies have leveraged advanced statistical models and ML classifiers to improve diagnostic accuracy from EEG and CDT datasets. For instance, Kim et al. developed the EEG-ML algorithm in order to detect brain amyloid beta pathology [6]. Ieracitano et al. have proposed that a convolutional neural network (CNN) approach could provide better EEG classification performance compared to other conventional learning machines. The average accuracy they achieved varies from 83.3% to 89.8% with CNN_1_, depending on which classification method they used—three-way (AD vs. MCI vs. HC) or binary (average of AD vs. MCI, AD vs. HC, and MCI vs. HC), respectively, whereas the epoch-based test results of CNN_2_ showed accuracies of 78.5% and 86.9% [7]. Hjorth parameters were also used in EEG signal analysis to improve the AD detection accuracy from 73.80% to 82.68% (SVM used) [8].

In the meantime, the clock drawing test has also been considered as a dataset to train deep neural network (DNN) models to detect AD patients. Regarding CDT, Chen et al. presented an automatic assessment of CDT with the aid of DNN [9]. Sato et al. also used a CDT-based DNN model to establish an automatic screening tool for detecting cognitive decline. The trained DNN model achieved an accuracy of 77.2% for discriminating probable dementia from HC [10].

Furthermore, some attempts have been made to utilize ensemble methods when analyzing some clinical datasets. Nanni et al. suggested an ensemble of classifiers for the early diagnosis of AD from pre-processed sets of T1-weighted MRI [11]. Chatterjee et al. ensembled multiple algorithms to accomplish a better predictive performance in a quick diagnosis of AD using brain longitudinal MRI data [12]. An et al. showed a deep ensemble learning framework using deep learning algorithms to merge multisource clinical datasets such as clinical evaluations, brain MRI, and neuropathology [13]. Moreover, Huang et al. have potentially overcome the limitation of three-dimensional patch CNN by introducing Monte Carlo ensemble vision transformer (MC-ViT), which employs a single vision transformer learner, instead of using traditional ensemble methods that deploy multiple learners [14].

As shown above, many machine learning model strategies have been established to improve the accuracy of disease detection. Ensemble learning (EL) is one of them, which utilizes multiple ML algorithms to achieve better predictive performance than that produced by a single algorithm [15]. Disease prediction models for diabetes, skin cancer, kidney disease, liver disease, and heart conditions have been suggested in an attempt to enhance disease detection rates [16]. However, few studies have utilized ensemble methods in the field of differential diagnosis of AD, as most researchers have focused on using EL for more prominent and extensively studied diseases, such as cancer, rather than neurodegenerative diseases. Moreover, the application of EL to AD classification has primarily been limited to using an ensemble of multiple classifiers that extract features from MRI datasets so far [11,12].

Instead of ensembles of probabilities calculated from multiple classifiers using a single data type, ensembles of probabilities calculated from multiple data types (in this study, EEG and CDT data) using a single classifier may provide a more suitable approach for the AD classification task. This is because a single base learner, or even ensembles of the base learners from a single data type (traditionally MRI data), often struggles to classify ambiguous cases that lie on the borderline between AD and HC. Such cases can be more effectively classified when the single data type is integrated with other data types. To leverage the synergies of combining multiple data types, multimodal machine learning approaches or ensemble learning methods are highly likely to represent a promising avenue. Yu et al. attempted to integrate features extracted from EEG, genotype and polygenic risk score data to obtain a better predictive performance [17]. Our study ensembled probabilities calculated from EEG and CDT features by a soft-voting method.

In this research, we propose that application of ensemble learning (EL) in data analyses of electroencephalogram (EEG) and clock drawing test (CDT) (Figure 1) can improve accuracies of detecting Alzheimer’s disease, compared to when machine learning is independently used in data analyses of either EEG or CDT, and can be used as an automatic diagnostic tool for classification of AD.

## 2. Materials and Methods

As the age of the subjects increases, so does the ratio of those diagnosed with AD according to our demographic analysis. Moreover, in cases of significantly high body mass index (BMI) above 30, the ratio of those diagnosed with AD was also high. It has been observed that higher midlife BMI increased the risk of dementia in long-term follow-up [18].

The overall experimental method is presented as a flowchart in Figure 2 below. We used three ML models with validated generalizability, and their utility has already been demonstrated through flexible application to detection and classification in numerous studies. Machine learning algorithms implemented in this study were random forest (RF) [19], extreme gradient boosting (XGB) [20], and support vector machine (SVM) [21]. The application of such ML algorithms training on EEG and CDT data was not limited to AD classification. RF has been used in sleep spindle detection [22] and in the classification of alcohol use disorder [23] based on EEG data. Davoudi et al. classified non-dementia and AD/vascular dementia patients by training RF on CDT data [24]. XGB has also been applied not only to AD classification, but to a variety of classification tasks, such as idiopathic generalized epilepsy [25], demonstrating higher disease prediction accuracy. SVM is a strong tool for disease classification tasks by generating hyperplane as long as the overfitting issue is resolved [26]. SVM has been used to classify movement intention from premovement EEG signals [27], and to analyze motor imagery EEG signals [28]. It has also been utilized for AD screening using the qualitative drawing error of CDT [29].

After extracting features from EEG signals and CDT images, random forest (RF) [19], extreme gradient boosting (XGB) [20], and support vector machine (SVM) [21] independently performed feature selection through analysis of variance (ANOVA), variance inflation factor (VIF), and recursive feature elimination (RFE). SHAP was used to determine how much each feature contributed to the classifier’s result prediction and judgment. Through this process, the three classifiers each selected 6 EEG features and 6 CDT features for use in AD diagnosis. The results predicted by each model using EEG and CDT data were classified into healthy control and AD through soft voting of ensemble learning (EL).

### 2.1. Data Collection

#### 2.1.1. Materials

This study utilized the dataset from Artificial Intelligence Learning Data Construction Project. Due to the retrospective nature of the study, the Institutional Review Board (IRB) of Gachon University Gil Medical Center, Incheon 21565, Republic of Korea (IRB Number: GDIRB2021-191) waived the need for obtaining informed consent. All experimental protocols were performed in accordance with the relevant guidelines and regulations in compliance with the Declaration of Helsinki. Data collection including recruitment of the subjects, clinical data collection, anonymization, testing, and applying to IRB for approval, was assigned to Dong-A University Hospital, Busan 49201, Republic of Korea and Pusan National University Hospital, Busan 49241, Republic of Korea. Onycom, Seoul 04335, Republic of Korea, was responsible for model performance evaluation (data setting for evaluation, data validation) and quality control (secondary testing, tertiary testing). SCT, Busan 48059, Republic of Korea, was in charge of data collection (cloud worker recruitment and management, brain disease integrative data collection) and administrative support. iMediSync Inc., Seoul 06247, Republic of Korea, also participated in collecting data (data collection processing and cloud environment setup), process brainwave data (EEG data collection, processing, and refinement), and conduct AI modeling (AI model planning, development, and validation). Lastly, SNC, Busan 46269, Republic of Korea, developed a CDT application.

Subjects who were positively and negatively screened were selected. The subjects were asked to get enough sleep and to not drink alcohol the day before the test day, and they were also asked to not consume any food or drink containing caffeine on the test day. The subjects were then placed in a quiet room where external noise was blocked. iSyncWave, a complete wireless dry EEG recording apparatus that consists of 19 EEG channels and 1 Fpz ground channel developed by iMediSync, was used as an EEG recording device. EEG was recorded in the resting state first, and then EEG was recorded while CDT was conducted. The data were collected from 19 scalp electrodes attached based on the International 10/20 System (Fp1, Fp2, F3, F4, F7, F8, Fz, C3, Cz, C4, T3, T4, T5, T6, P3, Pz, P4, O1, O2), and the sampling frequency was 250 Hz. A1 and A2 electrodes, which are linked-ear reference electrodes, were omitted. For the CDT, the subjects were asked to perform the test with a smart digital pen and a paper, and the whole process was recorded. Two medical specialists diagnosed the participants as either AD or HC based on the test results.

#### 2.1.2. Electronic Medical Record (EMR) Data Collection

The subjects were selected and screened according to a case report form including age, sex, years of schooling, diagnosis, cognitive test score, medication, CDR, DSM-V, and NIA-AA criteria. Patient information in the dataset was de-identified to preserve the privacy of the subjects, and encrypted IDs were newly granted.

Data with missing values were excluded from learning, the HC dataset was downsampled to resolve a class imbalance problem. In total, 199 cases of AD and 200 cases of HC were finally used in ML (Table 1).

### 2.2. Data Preprocessing

Both EEG and CDT data were preprocessed as part of the Artificial Intelligence Learning Data Construction Project. The data were preprocessed in accordance with the project’s collection guidelines below. These preprocessed datasets are accessible via the open-source AI Hub database upon request, and we employed the datasets to conduct post-preprocessing steps in this study.

#### 2.2.1. EEG Signal Preprocessing

Raw EEG signals underwent the following process to extract EEG features (Apparatus: iSyncBrain developed by iMediSync Inc.):Removal of transient noise introduced by movements during EEG measurement.Elimination of typical noise continuously introduced, based on frequency spectrum and topomap analysis.Exclusion of cases where the length of data extracted through the EEG refinement tool is less than 30 s, as it is difficult to ensure representativeness of individual resting-state EEG in such cases.

#### 2.2.2. CDT Image Preprocessing

CDT images also underwent the following process to extract CDT features (Apparatus: CVAT and Paint tools developed by SNC Co., Ltd.):Sending the test results conducted with the CDT application at each institution to the server.Image and video generation from the transmitted .json files.Extraction of the CDT performance timeline and coordinate values from the CDT app, and performance videos creation based on the timeline and coordinate values.Exclusion of some collected data if the timeline and coordinate values do not meet inclusion criteria.

### 2.3. Feature Extraction and Selection

Feature extraction and selection were performed independently for each classifier (RF, XGB, SVM). As a result, quantitative preprocessing of raw EEG signals and raw CDT images yielded a total of 21,440 EEG features and a total of 11 CDT features. EEG features were extracted and went through standardization. Features that had a statistically insignificant correlation with the labels (*p*-values greater than 0.05), as determined by ANOVA testing on the standardized data, were excluded [30,31,32]. This test was performed with one of the features as the independent variable and the presence of AD as the dependent variable. Through this process, 16,937 features were removed, leaving 4503 features. A significant number of features showed multicollinearity, which harmfully affected the performance of the classification model. To reduce such a multicollinearity problem, feature selection was conducted through recursive feature elimination (RFE), an algorithm that recursively eliminates the least important features from the dataset and repeats such an eliminating process until it reaches the desired model performance. For SVM, the calculations were based on weights, and for tree models (RF, XGB), the calculations were based on feature_importances [33]. This process helped reduce 4503 features to 200. After selecting 200 feature candidates through RFE, the feature having the highest variance inflation factor (VIF) was removed, and this procedure was repeated until VIF of all features reached below 5.(1)$$VIF_{i}=\frac{1}{1-R_{i}^{2}}$$

$VIF_{i}$ is a reciprocal of $1-R_{i}^{2}$, where $R_{i}^{2}$ is a coefficient of determination that measures how much the *i*-th independent variable is correlated to the remaining ones. The higher the $VIF_{i}$, the higher the possibility that multicollinearity exists. As mentioned above, any independent variables having $VIF_{i}$ higher than 5 were removed, leaving the least correlated variables. Depending on which algorithm (classifier) was used, $VIF_{i}$ also helped reduce the number of features to 113 for RF, 143 for XGB, and 152 for SVM. Each classifier re-conducted RFE, and the most valid 6 EEG features to be used in the classification step were selected. In contrast, only 11 features had initially existed in the CDT dataset, and 5 of them were removed through RFE. A total of 6 features from EEG and 6 features from CDT were finally selected (Table 2) by each classifier (RF, XGB, SVM), resulting in a total of 36 features.

Table 2 lists which EEG and CDT features were selected by each classifier and ranks the contribution of each feature set to the classification from highest to lowest. For RF, the most significant EEG feature was 2D_map_1hz_rel_7hz-O2 (relative average of 7 Hz band in the right occipital electrode), followed by TBR_value-T3 (theta/beta ratio in the left temporal electrode), conn_gamma_RAC_L-PC_R (functional connectivity in the gamma band between right Precentral and left Rostral Anterior Cingulate), conn_gamma_lsth_L-RAC_L (functional connectivity in the gamma band between left Rostral Anterior Cingulate and left Isthmus), conn_gamma_Fus_R-PC_R (functional connectivity in the gamma band between right Precentral and right Fusiform), and TBR_value_Fz (theta-beta ratio in the frontal electrode). For CDT features, the qualitative total score was the most significant, followed by needle labeling and order, numbering and order, quantitative total score, spatial and planning deficits, and conceptual deficits.

For XGB, the most significant EEG feature was conn_gamma_TP_L-CAC_R (functional connectivity in the gamma band between right Caudal Anterior Cingulate and left Temporal Pole), followed by 2D_map_1hz_rel_7hz-T3 (relative average of 7 Hz band in the left temporal electrode), 2D_map_1hz_rel_6hz-T6 (relative average of 6 Hz band in the posterior right temporal electrode), 2D_map_1hz_rel_8hz-O2 (relative average of 8 Hz band in the right occipital electrode), conn_gamma_lsth_R-RAC_L (functional connectivity in the gamma band between left Rostral Anterior Cingulate and right Isthmus), and 2D_map_1hz_abs_10hz-Cz (absolute average of 10 Hz band in the central midline electrode). For CDT features, the qualitative total score was the most significant, followed by spatial and planning deficits, needle labeling and order, numbering and order, clock face completeness, and pacing.

For SVM, the most significant EEG feature was 2D_map_1hz_rel_6hz-T3 (relative average of 6 Hz band in the left temporal electrode), followed by conn_gamma_PstC_L-RAC_R (functional connectivity in the gamma band between right Rostral Anterior Cingulate and left Postcentral), conn_gamma_PstCing_R-MOrb_L (functional connectivity in the gamma band between right Posterior Cingulate and left Medial Orbitofrontal), 2D_map_1hz_rel_7hz-T4 (relative average of 7 Hz band in the right temporal electrode), conn_beta2_Ent_L-TP_L (functional connectivity in the beta2 band between left Temporal Pole and Entorhinal), and conn_beta3_IP_L-Ins_R (functional connectivity in the beta3 band between right Insula and left Inferior Parietal). For CDT features, needle labeling and order was the most significant, followed by qualitative total score, spatial and planning deficits, numbering and order, watch size, and clock face completeness (Table 2).

### 2.4. Machine Learning and Classification

At the classification steps, we reutilized the same model we had used at the feature selection to optimize the final learning process (classification step). This is because selected features from a specific classifier during the RFE process are the best fit to itself when conducting a classification. Due to the small number of data samples, it is difficult to validate the generalization performance with just one test. Therefore, a 5-fold test was performed, and the standard deviation of each result was calculated as well (Table 3). We initially attempted to differentiate AD using 6 features each from EEG and CDT (① and ② in Figure 2) in order to compare with an ensemble (③ in Figure 2). The final result was determined through the ensemble using a soft voting method, where the prediction probabilities from models trained separately on CDT and EEG were added together to decide the presence of AD (③).

## 3. Results

A confusion matrix, a specific table layout to examine how much the labels predicted by the ensemble model match the true labels, was employed to visually represent the ensemble model’s performance (③ in Figure 2 and Figure 3). Each row represents the true class, and each column represents the model’s predicted class. Precisions, recall, F1-scores, and accuracy were calculated for each model (the rightmost side of Table 3). The RF model detects AD with considerably high accuracy, but it tends to misdiagnose some normal cases as AD more than other classifiers, which could be observed as the relatively high number of false positives (48, bottom left of RF in Figure 3). In the case of XGB, it achieves relatively higher accuracy for AD compared to other models, making it good at detecting AD, and it also contributes to reducing the misdiagnosis rate of normal cases with lower false positives (28, bottom left of XGB in Figure 3). Notably, the ensemble model shows stable performance with the highest AD detection accuracy (0.864, Table 3). On the other hand, SVM records the highest precision (0.972, Table 3) and shows excellent overall classification performance, but it has a higher false positive rate than that of XGB (35, bottom left of SVM in Figure 3), indicating a relatively higher likelihood of misdiagnosing normal cases as AD. Overall, the model applying ensemble learning demonstrates balanced performance in AD detection and normal case differentiation, yielding the best results.(2)$$Precision=TP/\left(TP+FP\right)$$(3)$$Recall=TP/\left(TP+FN\right)$$(4)$$F_{1}\;score=\frac{2\times Precision\times Recall}{Precision+Recall}$$

Note: *TP*, *TN*, *FP* and *FN*, respectively, stand for true positive, true negative, false positive, and false negative.

To visualize the model’s results, we plotted the receiver operating characteristic (ROC) curve and calculated the area under the curve (AUC) value as shown in Figure 4 and Table 4. As mentioned above, to quantify each model’s performance, we calculated the precision, recall, F1 score, and accuracy for each fold, and then we calculated the average and standard deviation of each metric from the multiple folds, shown in Table 3. Particularly, we also analyzed the performance of individual EEG and CDT models (Table 3, ① and ② in Figure 2) to compare with the effectiveness of the ensemble (Table 3, ③ in Figure 2).

For the Random Forest classifier, the statistical test results between the prediction values of EEG model and ensemble model, and the results between the prediction values of CDT model and ensemble model showed *p*-values of 0.875 and 0.885, respectively. As the *p*-values are greater than 0.05, the differences in prediction values between models were not statistically significant.

For XGB, the statistical test results between the prediction values of the EEG model and ensemble model, and the results between the prediction values of CDT model and ensemble model showed *p*-values of 0.929 and 0.922, respectively. As the *p*-values are greater than 0.05, the differences in prediction values between models were not statistically significant.

For SVM, the statistical test results between the prediction values of the EEG model and ensemble model, and the results between the prediction values of the CDT model and ensemble model showed *p*-values of 0.819 and 0.812, respectively. As the *p*-values are greater than 0.05, the differences in prediction values between models were not statistically significant.

McNemar’s Test was further conducted to statistically compare the accuracies between models. Three independent tests were performed because three models were trained on the classification task in total (_3_C_2_). Since three pairs were compared (RF vs. SVM, RF vs. XGB, SVM vs. XGB), the Bonferroni correction was applied to control for Type I error resulting from multiple comparisons. Based on the model comparison results in Table 5, the RF model showed significantly lower performance in terms of accuracy and AUC compared to the SVM and XGB models (*p* < 0.01). On the other hand, there was no statistically significant difference in performance between the SVM and XGB models for both accuracy and AUC (*p* > 0.2).

As a result, regardless of which classifier was used, the ensemble model demonstrated higher accuracy, F1 scores, precision, and recall compared to when using EEG and CDT features independently (Table 3). Notably, when using XGBoost (XGB) in ensemble learning (EL), it achieved significantly high scores with an F1 score of 0.863 and accuracy of 0.864 (Table 3). This suggests that the ensemble alleviated biases in false negatives and false positives, and it improved overall prediction performance. In conclusion, XGB emerged as the best-performing model, achieving the highest accuracy (0.864) and demonstrating the most significant improvement in accuracy compared to models using EEG (0.791) or CDT (0.771) alone.

SHAP (Shapley Additive Explanations) analysis was used to evaluate the explainability of the model results shown in Figure 5. SHAP analysis quantified the contribution of each feature in the model to the final prediction result, allowing us to analyze how and why the model produced these results [34]. It was analyzed that neural activity variables measured in specific brain regions had a significant impact on the model’s predictive performance.

Various features emerged as key characteristics in EEG, with the most significant features for each classifier being as follows:For RF: The relative average value of the 7 Hz frequency band in the 2D map of the O2 electrode (2D_map_1hz_rel_7hz-O2);For XGB: The functional connectivity in the gamma wave band between the right Caudal Anterior Cingulate and left Temporal Pole (conn_gamma_TP_L-CAC_R);For SVM: The relative average value of the 6 Hz frequency band in the T3 electrode (2D_map_1hz_rel_6hz-T3).

In CDT, for tree-based classifiers (RF and XGB), the most significant feature was the qualitative total score. In contrast, for SVM, the score for needle labeling and order was the most significant feature.

The ensemble model used a total of 12 features, 6 each from EEG and CDT features, with only the top 6 features displayed. The ensemble model showed that CDT features were more dominant than EEG features.

## 4. Discussion

Electroencephalography (EEG) and the clock drawing test (CDT) have long served as cornerstone conventional techniques for screening Alzheimer’s disease (AD), each offering unique insights into cognitive function and neural integrity. EEG is a noninvasive and cost-effective tool that records the brain’s electrical activity, providing real-time reflection of synaptic and neuronal function [35,36]. The application of machine learning techniques to EEG signal data has proven highly effective in the detection and classification of a wide range of neurological diseases, such as alcohol-related brain impairment [23], epileptic seizures, [37] and AD. The use of ML for AD classification has become an emerging trend in the field of biomedical engineering.

In AD, EEG analysis can reveal characteristic changes such as a shift toward slower brainwave frequencies, reduced complexity of neural signals, and decreased synchronization between brain regions—features that are increasingly leveraged as biomarkers for early detection and diagnosis [36]. Recent advances have incorporated computational and machine learning techniques to analyze EEG signals, extracting quantitative features such as spectral power, complexity, and synchronization, which have demonstrated high accuracy in distinguishing AD patients from HC [36,38,39].

Similarly, the clock drawing test (CDT) is a widely used neuropsychological assessment that evaluates visuospatial abilities and executive function, both of which are commonly impaired in AD. Machine learning algorithms trained on quantitative metrics derived from CDT performance—such as the spatial arrangement of numbers and hands—have also achieved notable success in detecting AD, complementing EEG-based approaches.

Figure 1 highlights CDT cases that were readily classified by these algorithms. A typical AD patient (AD–Easy case in Figure 1), when instructed to “draw a clock,” instead wrote the Korean words ‘시계’ (“clock”) and ‘11시 10분’ (“eleven-ten” or 11:10), rather than correctly arranging the numbers and clock hands. This type of response reflects a profound misunderstanding of the task, likely stemming from cognitive decline and impaired executive function. Such errors result in significantly lower scores across multiple CDT scoring systems, underscoring their diagnostic relevance [40]. The misinterpretation of instructions, as seen here, is a hallmark of cognitive deterioration in AD. On the other hand, individuals who accurately depict a clock face with the numbers correctly positioned and the hands set to the requested time typically demonstrate preserved cognitive abilities.

Integrating CDT-derived visuospatial metrics with EEG signal-derived quantitative metrics demonstrates complementary effectiveness by leveraging distinct measurement modalities, thereby not only enhancing detection accuracy but also providing a more nuanced understanding of the diverse manifestations of AD, paving the way for earlier and more reliable detection [41,42]. In this study, the ensemble approach combining such two data sources, which is a method that has not been taken, achieved improved classification performance compared to individual use of either modality alone. Feature extraction and selection helped identify six highly significant features from the EEG signals, which were consistent with findings from previous studies; it was reported that altered ‘functional connectivity in the gamma band’ [17,43] and altered ‘theta-beta power ratio’ in AD patients were observed [44]. Similarly, six highly significant features selected from the CDT images were also consistent with findings from previous studies, such as ‘numbers in the current position’, ‘minute target number indicated,’ and ‘hand in correct proportion’ [29,45].

### 4.1. Role and Strength of Ensemble Methods in the Alzheimer’s Disease Classification Task

There are a couple of reasons why EL is more than qualified for the AD classification task. First, conventional supervised learning algorithms based on a single model (a base learner) can achieve significantly high accuracy at a decent level as well, but they often struggle with hard-case classifications. In contrast, an ensemble of two distinct data sources—EEG signal-derived quantitative data and CDT-derived visuospatial data—mitigates this issue through soft voting, as it relies on the concept of diversity [15]. For instance, in a case when the CDT classifier with hard-to-classify CDT data predicts that the CDT image is of an AD with a probability of 0.5 (refer to ‘Ensemble’ box in Figure 2), it is prone to mistakes. On the contrary, the ensemble of such a probability score with an AD probability of 0.8 predicted by the EEG classifier yields a relatively more confident prediction, resulting in a total probability of 0.65. Second, EL can also effectively integrate multiple modalities during high-dimensional data analysis [17]. The combined use of EEG and CDT features provides complementary insights into functional brain changes associated with AD.

In this study, the soft voting ensemble of calculated probabilities from EEG and CDT classifiers consistently outperformed individual EEG and CDT feature-based models across all tested classifiers, showing superior results in accuracy, F1 scores, precision, and recall (Table 3). XGB within the ensemble framework delivered the strongest performance, achieving an F1 score of 0.863 and accuracy of 0.864 (Table 3). These results indicate the ensemble approach effectively alleviated prediction biases by reducing both false positives and false negatives. XGB emerged as the most effective model, attaining the highest accuracy (0.864) and showing marked improvement over standalone EEG (0.791) or CDT (0.771) implementations.

### 4.2. Appropriateness of Machine Learning Models Chosen for Training on EEG and CDT Data

Machine learning algorithms employed in this study were random forest (RF), extreme gradient boosting (XGB), and support vector machine (SVM). These three algorithms applied in this study are all supervised learning methods, which generally achieve higher accuracy on average in EEG classification tasks compared to their unsupervised counterparts [15].

RF is an algorithm that constructs decision trees using the bagging method. By generating diverse sub-datasets through bootstrapping and training multiple decision trees on their respective datasets, it aggregates their results to alleviate the overfitting issues common in single decision trees. RF has proven to be effective in handling high-dimensional and multi-source data, such as that used in this study [46].

XGB is a distributed, open-source machine learning library that employs gradient-boosted decision trees, a representative supervised learning boosting algorithm utilizing gradient descent. While RF is based on the concept of bagging (bootstrap aggregation), where individual trees are trained independently and their predictions combined, boosting algorithms such as XGB use an additive approach, training weak learners sequentially to correct errors from prior models. Such a combining method in an additive manner computes predictions 10 times faster than RF [47]. Notable advantages of XGB include efficient model tuning, high scalability, algorithmic optimization, and low computational load [48,49,50,51]. As a result, this algorithm has continuously demonstrated strong predictive performance in analyzing EEG signals and other features from patients with neurological diseases [17,25,52].

SVM is also a supervised machine learning algorithm that classifies data by identifying an optimal line or hyperplane in N-dimensional space to maximize the distance between classes. It is one of the most widely used algorithms for EEG analysis in both classification and regression tasks. Specifically, it employs kernel functions (“kernel trick”) to address challenges in linearly separating complex data by transforming lower-dimensional data into higher-dimensional space, enabling linear classification [15,53]. SVM is also a valuable tool for image classification and segmentation, as well as hand-written character recognition, having implied its usage in CDT image analysis [54,55,56].

### 4.3. Role of Analysis of Variance (ANOVA) in Feature Selection

Analysis of variance (ANOVA) is a statistical test widely used to compare and contrast the averages of two or more groups by analyzing variance. ANOVA does not necessarily provide predictive power, and applying it without correction may lead to false positive results, making it potentially inappropriate. However, ANOVA was primarily used for dimensionality reduction due to the large number of features (21,440). To consider predictive performance, RFE was utilized, and VIF was calculated to eliminate multicollinearity and redundancy. Since calculating VIF is computationally expensive, RFE was performed both before and after VIF calculation to further reduce dimensions.

The main reason for using ANOVA was to reduce computational costs, as ANOVA (O(n)) has relatively lower computational costs compared to RFE (O(n^2^)) and VIF (O(n^3^)). ANOVA may not be an appropriate tool for evaluating predictive performance indeed. However, applying RFE and VIF calculations directly to features with 21,440 dimensions was burdensome. Some of the previous studies have used ANOVA as a tool to dramatically reduce the number of features by minimizing the computational burden [30,31,32]. Moreover, Kim et al. employed ANOVA to extract and select EEG features in order to examine whether pre-movement EEG signals could be used to decode parameters like direction, distance, and target positions in reaching tasks [57]. Although ANOVA alone does not guarantee predictive usefulness, in this study, its role was strictly limited to the initial stage of dimensionality reduction. After that, predictive relevance was thoroughly evaluated through subsequent feature selection based on models (RFE) and multicollinearity control (VIF). Finally, the most beneficial features were isolated through a final RFE after evaluating predictive performance.

### 4.4. Other Strengths and Limitations

Ensemble learning could make progress in detecting AD compared to using EEG and CDT data independently in machine learning algorithms, but there are still some points to be improved in the future. First, in ambiguous cases, there were occasional instances of AD being classified as normal (false negative) or normal cases being classified as AD (false positive). For early diagnosis of dementia, it is crucial to accurately differentiate these hard cases, as improving the diagnostic success rate for such ambiguous patient groups is essential.

Traditional scoring systems of CDT such as Shulman’s (5-point scale) and Sunderland’s (10-point scale) are the conversion of qualitative image data into quantitative numerical values, which are manually estimated and performed by the clinicians. Further analysis is therefore required in terms of setting up the automated quantitative scoring system of CDT.

More specifically, the CDT scoring dataset presented in this paper is manually estimated by the clinicians. It is therefore, in the future, necessary to establish a fully automated scoring algorithm in order to avoid the bias made by an individual clinician’s decision. Regarding EEG data analysis, further study with up-to-date Multi-Layer Perceptron may be needed because in this paper conventional ML model has been used in classifying the dataset. Furthermore, there are basically still aspects of the model’s performance that have not been clinically verified. In other words, though the model may show promising results in a research or experimental setting, it has not yet undergone rigorous clinical testing or validation. This is an important caveat in medical research, particularly in the development of diagnostic tools or predictive models for conditions like AD.

With regard to CDT, its inherent relevance to motor control provokes potential confounding effects of non-AD-related motor impairments on test outcomes. Even though this study excluded participants with neurological diseases affecting motor function (e.g., Parkinson’s), individual variations such as baseline drawing proficiency and handgrip muscular strength (HGS) were not thoroughly considered. These factors warrant further exploration, as prior research has demonstrated significant associations between CDT performance and such measures. For example, Fastame et al. reported that handgrip strength and functional mobility accounted for 12–19% of the variance in CDT results, highlighting the need to disentangle motor and cognitive contributions in future analyses [58].

VIF stands for Variance Inflation Factor, which is a measure used to detect multicollinearity in regression analysis. A VIF of 1 means there is no correlation between a given feature and any others, whereas VIFs above 5 or 10 (depending on the threshold used) typically indicate problematic levels of multicollinearity. A lot of features seemed to have multicollinearity, which negatively affected the performance of the classification model. This issue was mitigated through a process of measuring the VIF after selecting 200 feature candidates, and then removing the features with the highest VIF until all features had a VIF below 5.

Furthermore, although CDT and EEG are sets of data correlated to each other, CDT and EEG datasets were independently applied to EL without any analysis regarding their correlation. Therefore, it is needed for future research endeavors to invent correlative analysis methods to further improve disease detection accuracy.

## Figures and Tables

**Figure 1 sensors-25-02881-f001:**
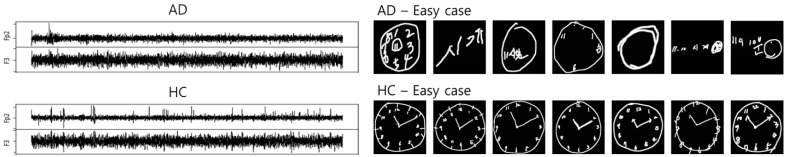
Representative EEG and CDT data examples used in this study. (Left) Examples of EEG signals. This figure demonstrates datasets sampled at 250 Hz and cut to a length of 4 s, partially showing electrical signal collected from 2 channels (Fp2, F3) out of a total of 19 channels as examples. Such acquired signal data were processed to generate a variety of EEG features. (Right) Typical easy-to-classify cases from CDT images. The subjects were asked to draw a clock face, place the numbers, and set the hands to a specific time.

**Figure 2 sensors-25-02881-f002:**
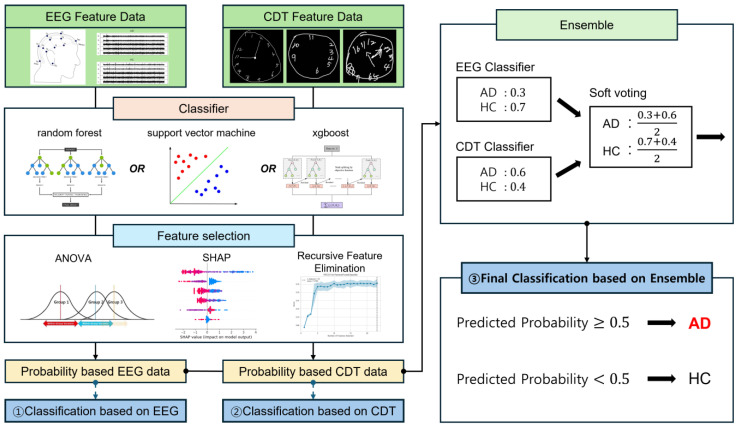
Flowchart—an overview. A schematic of how each EEG and CDT dataset was analyzed. EEG and manually scored CDT data are (1) commonly processed into feature data, (2) three classifiers independently perform feature selection to sort out only a small number of probability-based EEG and CDT feature datasets, and (3) these selected EEG and CDT feature datasets are either directly used to classify AD (① and ②) or ensembled through soft voting to infer the final result (③).

**Figure 3 sensors-25-02881-f003:**
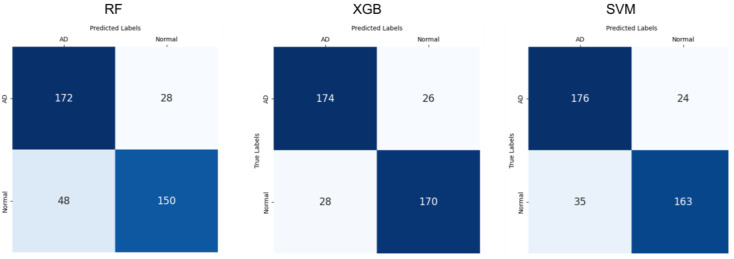
Confusion matrices of EL of Random Forest (RF), eXtreme Gradient Boosting (XGBoost), and Support Vector Machine (SVM) for comparison.

**Figure 4 sensors-25-02881-f004:**
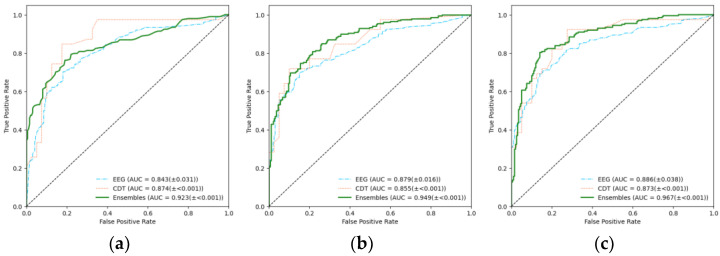
ROC curves for HC and AD from the (**a**) RF, (**b**) XGB, and (**c**) SVM classifiers, presented from left to right.

**Figure 5 sensors-25-02881-f005:**
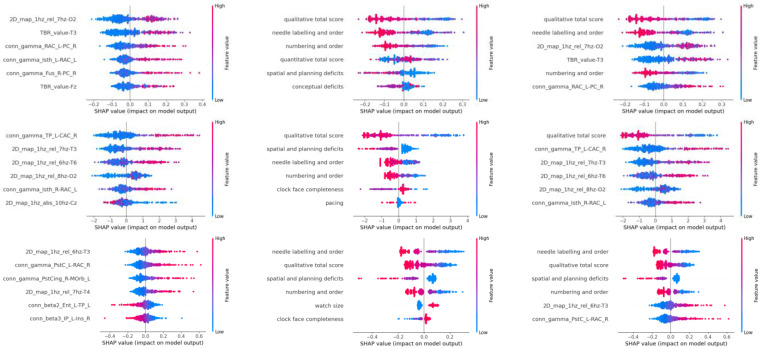
Results of SHAP analysis. From left to right: EEG only, CDT only, ensemble; from top to bottom: RF, XGB, SVM classifiers. The redder the color, the higher the feature value; the bluer the color, the lower the feature value. The further to the right, the more the feature influences a positive prediction; the further to the left, the more the feature influences a negative prediction. (e.g., If it transitions from blue on the left to red on the right, it indicates a positive correlation; if it transitions from red on the left to blue on the right, it indicates a negative correlation.) The abbreviations for brain regions shown in the graph are as follows, RAC_L: Rostral Anterior Cingulate (left); RAC_R: Rostral Anterior Cingulate (right); CAC_L: Caudal Anterior Cingulate (left); CAC_R: Caudal Anterior Cingulate (right); TP_L: Temporal Pole (left); TP_R: Temporal Pole (right); MOrb_L: Medial Orbitofrontal (left); MOrb_R: Medial Orbitofrontal (right); Ins_L: Insula (left); Ins_R: Insula (right); PstC_L: Postcentral (left); PstC_R: Postcentral (right). The abbreviations for values are as follows, TBR: Theta-Beta Ratio, conn_(frequency band; TBR): brain spatial connectivity in the frequency band; 2D_map: feature values by frequency band and sensor; 2D_map_1h: feature values by 1 h band and sensor; abs: average; rel: relative average. For the ensemble model, the CDT item has high importance in all classifiers, with RF and XGB particularly utilizing CDT data well. For EEG, XGB and SVM show a more balanced feature usage tendency compared to RF (RF tends to focus on positive correlations).

**Table 1 sensors-25-02881-t001:** Number of training and test data for AD and HC. Due to the small amount of data, training was performed using 5-fold cross-validation. Depending on the fold, the number of data used in some cases is shown in parentheses.

Classes	Train	Test	Total
Alzheimer’s Disease	160 (159)	39 (40)	199
Healthy Control	160	40	200

**Table 2 sensors-25-02881-t002:** EEG and CDT features selected through ANOVA, RFE, and VIF. The higher the rank, the greater the contribution of each feature to the classification decision of each classifier.

Data	Rank	RF	XGB	SVM
EEG	1	2D_map_1hz_rel_7hz-O2	conn_gamma_TP_L-CAC_R	2D_map_1hz_rel_6hz-T3
	2	TBR_value-T3	2D_map_1hz_rel_7hz	conn_gamma_PstC_L-RAC_R
	3	conn_gamma_RAC_L-PC_R	2D_map_1hz_rel_6hz-T6	conn_gamma_PstCing_R-MOrb_L
	4	conn_gamma_lsth_L-RAC_L	2D_map_1hz_rel_8hz-O2	2D_map_1hz_rel_7hz-T4
	5	conn_gamma_Fus_R-PC_R	conn_gamma_lsth_R-RAC_L	conn_beta2_Ent_L-TP_L
	6	TBR_value_Fz	2D_map_1hz_abs_10hz-Cz	conn_beta3_IP_L-Ins_R
CDT	1	Qualitative Total Score	Qualitative Total Score	Needle Labeling and Order
	2	Needle Labeling and Order	Spatial and Planning Deficits	Qualitative Total Score
	3	Numbering and Order	Needle Labeling and Order	Spatial and Planning Deficits
	4	Quantitative Total Score	Numbering and Order	Numbering and Order
	5	Spatial and Planning Deficits	Clock Face Completeness	Watch Size
	6	Conceptual Deficits	Pacing	Clock Face Completeness

**Table 3 sensors-25-02881-t003:** Precision, recall, F1 scores, and accuracy of EEG, CDT, and ensemble from three classifiers (RF, XGB, SVM). Standard deviations are also provided next to each mean value.

Classifier	Metrics	EEG	CDT	Ensembles
RF	Precision	0.800 (±0.059)	0.853 (±0.043)	0.865 (±0.031)
	Recall	0.737 (±0.051)	0.702 (±0.110)	0.758 (±0.055)
	F1-Score	0.766 (±0.052)	0.740 (±0.071)	0.798 (±0.036)
	Accuracy	0.776 (±0.051)	0.759 (±0.053)	0.809 (±0.031)
XGB	Precision	0.848 (±0.042)	0.875 (±0.044)	0.917 (±0.049)
	Recall	0.773 (±0.059)	0.692 (±0.081)	0.859 (±0.041)
	F1-Score	0.786 (±0.035)	0.749 (±0.041)	**0.863 (±0.038)**
	Accuracy	0.791 (±0.032)	0.771 (±0.029)	**0.864 (±0.039)**
SVM	Precision	0.909 (±0.072)	0.824 (±0.053)	**0.972 (±0.085)**
	Recall	0.763 (±0.037)	0.708 (±0.134)	0.824 (±0.069)
	F1-Score	0.806 (±0.045)	0.754 (±0.083)	0.850 (±0.052)
	Accuracy	0.817 (±0.046)	0.777 (±0.061)	0.854 (±0.054)

**Table 4 sensors-25-02881-t004:** A table summarizing the AUC values of the ROC curves is shown below. The AUC values of ensemble learning are higher than those of either CDT or EEG alone across all classifiers.

	RF	XGB	SVM
**EEG**	0.843 (±0.031)	0.879 (±0.016)	0.886 (±0.038)
**CDT**	0.874 (±<0.001)	0.855 (±<0.001)	0.873 (±<0.001)
**Ensemble (EEG+CDT)**	**0.923 (±<0.001)**	**0.949 (±<0.001)**	**0.967 (±<0.001)**

**Table 5 sensors-25-02881-t005:** Statistical comparison of accuracy (*p*-values, McNemar’s test).

Model Comparison	*p*-Value	Statistical Significance (α = 0.016)
RF vs. SVM	0.0019	Significant difference
RF vs. XGB	0.0019	Significant difference
SVM vs. XGB	0.2200	No significant difference

## Data Availability

The preprocessed data utilized in this article were accessed on 4 December 2024 and are available in the AI Hub open-source database (https://www.aihub.or.kr/aihubdata/data/view.do?currMenu=115&topMenu=100&aihubDataSe=data&dataSetSn=71397). Other private data presented in this article are available on request from the corresponding author due to ethical reasons unless otherwise mentioned.

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
