# Peer review of "Ensemble Learning-Based Alzheimer’s Disease Classification Using Electroencephalogram Signals and Clock Drawing Test Images"

_sensors, 2025, doi:10.3390/s25092881_

Round 1
Reviewer 1 Report
Comments and Suggestions for Authors
There is no explanation of what features are used and how they are calculated. Are features calculated for each channel, or are features calculated between channels?
How long is the EEG signal data so the resulting feature is 21440?
There is no explanation of how many EEG features and how many CDT features
Recursive feature elimination (RFE) plays an important role in feature reduction, but RFE details are not given, and no references are given so that readers can understand the RFE process
Some of the methods used are not explained in detail, such as SHAP (Shapley Additive Explanations), which also does not provide adequate references
The discussion provided is just like reviewing the theory of using ML for AD detection; does not discuss the results obtained, why such results were obtained, and what the implications are
Reviewer 2 Report
Comments and Suggestions for Authors
- Introduction needs to be clearly specify the meaning of ensemble learning. Though, authors have provided the usual meaning of ensemble learning, it does not accurately reflect how it was implemented in the study. It is more of combining datasets (EEG vs CDT vs Combined EEG and CDT) and letting the machine learning model classify on each of these datasets. (Random Forest alone, vs SVM alone vs XGB alone). Apparently, the ML models can better a prediction if the EEG and CDT datasets were combined rather than singly. In other words, ML models perform better if we use combined or hybrid datasets.
- For the methods section, perhaps it would be clearer if description of the details (such as number of attributes, attributes datatypes etc) are also presented.
- I agree with the feature selection employed in the study. The rationale was clearly detailed in the methods section.
- For the evaluation of the performance, specify which metric shall be used to identify the best performing model. In their analysis, almost all metrics (accuracy, recall, precision, F1-score and AUC) improve with the combined EEG and CDT for the three ML models. Which ML model would then be chosen as the best performing model and specify the basis.
- It would also be better if comparison with existing literature for AD diagnosis was included in the discussion. In this way, the combined use of the datasets (EEG and CDT) would highlight its clinical utility in the diagnosis of AD. This could also lead the way for is application in other disease domains.
Comments on the Quality of English Language
Quality of English is fine.
Reviewer 3 Report
Comments and Suggestions for Authors
This study performed Alzheimer’s disease classification using ensemble learning applied to EEG signals. Their methods showed improved AD detection accuracy. While their methods may be valuable, I think the manuscript needs some improvement.
What is the figure in the introduction? It does not have 'Figure' in the caption.
The non-title figure is quite interesting. What exact instructions were given to the performers? Drawing a clock is understandable, but writing letters is particularly intriguing.
I felt the introduction lacks strong justification for the study. It simply states that EL has been less used in AD classification without clearly explaining why this is important or what specific gap this study addresses. The authors should clarify why existing EEG/CDT-based methods are insufficient and explicitly state how EL provides an advantage.
The authors did not specify the EEG recording device used.
Line 136: The meaning of 'previously classified 33 quantified data' is unclear. Clarify this.
I couldn’t find a description of the EEG preprocessing steps. EEG signals are highly sensitive to noise, so preprocessing is crucial. You should provide a full, step-by-step description of all preprocessing applied, including relevant parameters and tools, in chronological order, without omissions. This typically includes steps like filtering, re-referencing, resampling, inspections, channel and time-segment rejection, interpolation, and advanced artifact removal (e.g., ASR, ICA). Remember that you should provide enough information for readers to replicate your study. In general, EEG preprocessing is described in a separate dedicated subsection in the methods.
There is no justification for selecting RF, XGB, and SVM as your classification models. Why were these specific models chosen over other possible approaches? Model selection should be clearly justified rather than assumed. One way to justify this is by discussing the application of these methods in various EEG classification tasks, demonstrating their generalizability. For example, SVM has been used in motor tasks [1] and motor imagery [2]. Introducing such examples can strengthen your reasoning for selecting these models.
[1] https://doi.org/10.3389/fnhum.2019.00063
[2] https://doi.org/10.1155/2016/4941235
Line 154: The term '21,440 EEG feature datasets' is unclear. Did you really use 21440 separate datasets? It seems contradictory to the subsequent description.
The feature selection process is totally unclear, especially the role of ANOVA. It is not specified how ANOVA was applied. What was the dependent variable? Was a separate ANOVA performed for each EEG feature, comparing AD vs. HC? If so, this approach is problematic because statistical significance (p-value) does not necessarily indicate predictive power in a classification task. Also, given the large number of features (21,440), applying ANOVA without multiple comparisons correction (e.g., Bonferroni correction) could result in false positives.
Furthermore, ANOVA is not inherently a feature selection method; it only tests whether mean differences exist between groups, not whether a feature improves classification performance. Please clarify why ANOVA was applied before other selection methods (RFE, VIF) and how the results influenced feature elimination.
The model comparison lacks proper statistical validation. While classification metrics (like accuracy, F1-score) are reported, no statistical tests are used to determine whether performance differences between models are significant. Without statistical validation, the conclusions drawn from these results remain unreliable.
I don’t see any real discussion in 4.1 and 4.2. In 4.1, the explanation of EEG-based ML for AD diagnosis is too general and disconnected from your actual findings. Instead of discussing your results, you only provide generic statements about EEG’s limitations and the advantages of ML. There is no analysis of how your approach improves EEG-based AD classification, nor any comparison with prior ML-based EEG studies. What specific EEG features contributed most to your model? How did ensemble learning improve over traditional ML models? You need to provide a real discussion of your results, not just a broad claim that machine learning is better than conventional EEG analysis. Similarly, Section 4.2 has the same problem.
Also, lines 329-330 are misleading and conceptually illogical. EEG itself is a neurophysiological recording method, not a process that can be 'time-consuming' or 'subjective.' While EEG data analysis can be complex, the claim that machine learning 'improves EEG' does not make sense. ML can assist in analyzing EEG signals, but it does not change the fundamental nature of EEG itself.
Lines 331-333: Furthermore, while machine learning can enhance EEG analysis, it does not inherently reduce time consumption, as preprocessing and feature extraction often require additional computational steps. In addition to that, what exactly are you comparing? The phrase 'more efficient' is unclear. more efficient than what? EEG-based analysis has many variations, so without a clear point of comparison, this statement lacks meaning.
A discussion typically includes interpretations of findings, comparisons with previous studies in relevant contexts, practical implications, limitations, and future directions. Your claim is too broad and general. This section needs to be rewritten.
CDT inherently involves motor control, yet this aspect is not addressed in your discussion. Given that motor impairments can affect CDT performance, I expected you to interpret your results in light of this. How do your findings relate to motor control processes? I think it would be helpful to discuss the results in terms of this.
Round 2
Reviewer 1 Report
Comments and Suggestions for Authors
The author has revised this paper according to the reviewer's comments. The addition of several paragraphs clarifying the contents marks significant changes.
The reasons for choosing ML, the correlation between task and AD detection, and the selection of features are explained in full so that this paper is worthy of publication in this journal.
Author Response
The author has revised this paper according to the reviewer's comments. The addition of several paragraphs clarifying the contents marks significant changes. The reasons for choosing ML, the correlation between task and AD detection, and the selection of features are explained in full so that this paper is worthy of publication in this journal.
Response: I would like to express my sincere gratitude for the comments so far. Thanks to your comments, I was able to make the corrections successfully.
Reviewer 3 Report
Comments and Suggestions for Authors
I appreciated the response. However, some issues have not been fully addressed.
Comments 3: The non-title figure is quite interesting. What exact instructions were given to the performers? Drawing a clock is understandable, but writing letters is particularly intriguing.
Response 3: Thank you for pointing this out. I added some details about the exact instructions that were given to the subjects and modified some ambiguous expressions
- The authors clarified what was instructed but ignored the unexpected behavior (writing letters) that I found intriguing. Again, why did AD participants write letters instead of (or alongside) numbers? This part still needs to be explained.
Response 7: Thank you for pointing that out. I made a subsection in the methods section describing how the EEG and CDT data were preprocessed.
- The response is very unsatisfactory. The added part is insufficient. How was the noise removed? Was it automatic? Manual? Based on amplitude thresholds? Was ICA, ASR, or any other technique used? No mention.
- For "Elimination of typical noise continuously introduced, based on frequency spectrum and topomap analysis," again, this is very unspecific. Which frequencies were removed? Was it a notch filter? A band-pass? What kind of "typical" noise? How did you remove it? With what methods? What tools? What parameters?
- Again, you should provide a full, step-by-step description of all preprocessing applied, including relevant parameters and tools, in chronological order, without omissions. This typically includes steps like filtering, rereferencing, resampling, inspections, channel and time-segment rejection, interpolation, and advanced artifact removal (e.g., ASR, ICA). Remember that you should provide enough information for readers to replicate your study. In general, EEG preprocessing is described in a separate dedicated subsection in the methods.
- Refer to this paper (https://doi.org/10.1109/TNSRE.2025.3527578) for an example of how EEG processing can be described in a more complete and replicable way.
Response 8: I would like to express my sincere gratitude for this detailed comment. Thanks to this comment including some examples, I added a subsection in the Discussion section explaining why I selected those models.
- The revised subsection is not sufficient. I specifically asked the authors to demonstrate generalizability, but instead they just explained the plain methodology of each model, which is not what I asked. In addition to generalizability, they still do not justify their selection. I also provided examples of the use of SVM to help guide a concrete clarification and avoid confusion, but the authors did not refer to or engage with them, which is very disappointing. Again, explanations of how these models work belong in the methods, not in place of what I asked.
Response 11: I really appreciate this detailed comment. Thanks to the comment, I also added a paragraph to justify the use of ANOVA in feature selection with some previous studies that attempted to use ANOVA to reduce the dimensionality of features. âš« https://doi.org/10.1007/s11227-023-05179-2 âš« https://doi.org/10.1039/C4MB00316K âš« https://doi.org/10.1016/j.ymeth.2022.10.008
- I think these are good references. However, the three cited studies are not EEG-related or relevant to the current domain. At best, they show that ANOVA can be used for dimensionality reduction, but not that it’s appropriate or effective in this context. I encourage the authors to find more examples within the EEG domain. Fortunately, I’m aware of a previous study that used ANOVA for EEG feature extraction in a classification task (https://doi.org/10.3389/fnins.2019.01148). I believe this would be helpful, as it is one of the most direct and relevant examples.
Response 12: I added a table demonstrating the McNemar’s test results, and also added a paragraph regarding the table.
- I think the table is an important addition to the evaluation. However, it does not mention whether any correction for multiple comparisons was applied. Please address this as well.
Response 13: I revised the entire paragraphs and added a few paragraphs both before and after.
- The revised discussion is somewhat improved, but it still does not meet the expectations I outlined. The section remains mostly conceptual and hypothetical, without directly engaging with the results of this study. For example, no specific EEG features are mentioned, nor is there any quantitative discussion of how ensemble learning improved performance over EEG or CDT alone. I also requested a comparison with previous EEG+ML studies, which is still missing. Please revise again to provide a genuine discussion of your own findings.
